# Galectin-10 as a Potential Biomarker for Eosinophilic Diseases

**DOI:** 10.3390/biom12101385

**Published:** 2022-09-27

**Authors:** Hiroki Tomizawa, Yoshiyuki Yamada, Misaki Arima, Yui Miyabe, Mineyo Fukuchi, Haruka Hikichi, Rossana C. N. Melo, Takechiyo Yamada, Shigeharu Ueki

**Affiliations:** 1Clinical Laboratory Medicine, Department of General Internal Medicine, Akita University Graduate School of Medicine, Akita 010-8543, Japan; 2Department of Otorhinolaryngology, Head and Neck Surgery, Akita University Graduate School of Medicine, Akita 010-8543, Japan; 3Department of Pediatrics, Tokai University School of Medicine, Isehara 259-1193, Japan; 4Laboratory of Cellular Biology, Department of Biology, Institute of Biological Sciences, Federal University of Juiz de Fora, Juiz de Fora 36036-900, MG, Brazil

**Keywords:** eosinophil, ETosis, eosinophil extracellular trap, Charcot–Leyden crystals, galectin-10

## Abstract

Galectin-10 is a member of the lectin family and one of the most abundant cytoplasmic proteins in human eosinophils. Except for some myeloid leukemia cells, basophils, and minor T cell populations, galectin-10 is exclusively present in eosinophils in the human body. Galectin-10 forms Charcot–Leyden crystals, which are observed in various eosinophilic diseases. Accumulating studies have indicated that galectin-10 acts as a new biomarker for disease activity, diagnosis, and treatment effectiveness in asthma, eosinophilic esophagitis, rhinitis, sinusitis, atopic dermatitis, and eosinophilic granulomatosis with polyangiitis. The extracellular release of galectin-10 is not mediated through conventional secretory processes (piecemeal degranulation or exocytosis), but rather by extracellular trap cell death (ETosis), which is an active cell death program. Eosinophils undergoing ETosis rapidly disintegrate their plasma membranes to release the majority of galectin-10. Therefore, elevated galectin-10 levels in serum and tissue suggest a high degree of eosinophil ETosis. To date, several studies have shown that galectin-10/Charcot–Leyden crystals are more than just markers for eosinophilic inflammation, but play functional roles in immunity. In this review, we focus on the close relationship between eosinophils and galectin-10, highlighting this protein as a potential new biomarker in eosinophilic diseases.

## 1. Introduction

Several biologics are promising therapeutic options for severe eosinophilic inflammation, such as severe asthma. Elevated peripheral blood eosinophil counts, possibly reflecting airway eosinophilia, are a good indicator for determining biologic use [1,2,3]. As a clinical biomarker, the peripheral blood eosinophil count is useful for assessing eosinophilic inflammation. However, not only eosinophilic inflammation, but also various conditions, such as age, smoking history, and gender, may affect the peripheral blood eosinophil count [4]. Systemic immune reactions, such as systemic inflammatory-response syndrome and bacterial infection, can cause eosinopenia [5,6]. However, impaired production of endogenous corticosteroid, as observed in adrenal insufficiency [7] and malignancies, occasionally induce eosinophilia [8]. Even more problematic is that peripheral blood eosinophils represent only a small fraction of these cells, which likely remain unprimed in the blood. However, the majority of eosinophils reside in tissues [9] and can be activated in the process of recruitment and migration [10]. The peripheral blood eosinophil count is not always perfect for determining eosinophilic inflammation. Therefore, the direct observation of infiltrated eosinophils in the involved organs is often required. To improve the accuracy of diagnosis and disease severity, studies have been conducted, for decades, to identify blood biomarkers of eosinophilic inflammation.

## 2. Assessing Eosinophilic Inflammation

Eosinophil activation is characterized by the attraction of these cells to the target organ and the release of their bioactive products [11]. Eosinophils comprise a major population of secretory granules (specific granules) that contain preformed eosinophil-specific proteins (i.e., major basic protein-1, 2 [MBP-1, 2], eosinophil peroxidase, eosinophil cationic protein, and eosinophil-derived neurotoxin (EDN), as well as more than 30 cytokines [12,13]. The concentrations of these granule proteins in blood, tissues, and secretions might reflect direct biomarkers for assessing eosinophilic inflammation.

Classical studies have attempted to assess eosinophil activation with hyper-segmented nuclei-eosinophils or hypo-dense eosinophils [14]. Ex vivo cellular functions and surface proteins of peripheral blood eosinophils are considered to be surrogate markers for eosinophilic inflammation. The expression of adhesion molecules, such as CD11b [15] and CD49d [16], and primed reactive oxygen species production [17], survival [18], adhesion [19], and chemotaxis [20] can be assessed using peripheral blood eosinophils.

To assess tissue eosinophils, body fluids, such as sputum and bronchoalveolar lavage fluid, urine, nasal discharge, breath, and feces, as well as biopsied tissues, are available. In particular, an increased percentage of sputum eosinophils is correlated with severe diseases in asthma [21] and they were recently shown to be a marker for biologic treatments [22]. Urine [23] and feces [24,25] EDN can also be used for assessments of asthma and eosinophilic gastrointestinal disorders. In the clinical setting, exhaled fractional nitric oxide concentrations in the breath are used as a marker of airway eosinophilic inflammation [26]. 

When biopsy samples are available, histopathological analyses provide more meaningful information. In addition to routine staining, immunostaining and cell sorting are used in the research setting [27]. However, sampling, standardization of the methods, and evaluations are more difficult than those in peripheral blood. More importantly, the distribution of eosinophils is not even among organs. Physiologically, most organs lack resident eosinophils [28]. Some exceptions include the spleen, lymph nodes, and thymus tissue, which possess physiological eosinophils in the absence of degranulation. However, with the exception of the esophagus, eosinophils are normally found in the gastrointestinal tract with remarkable degranulation [28]. 

## 3. Abundance and Distribution of Galectin-10 in Human Eosinophils

Galectins are a family of lectins found in a wide range of species from sponges to humans [29]. Structurally, galectin is a carbohydrate-binding protein with specific β-galactosides and an approximately14-kDa carbohydrate recognition domain. Galectins are classified as galectin-1 to galectin-15 depending on the structure of the carbohydrate recognition domains. Galectins are involved in a plethora of biological phenomena, such as cell migration, autophagy, and signaling [30], immune responses, and diseases including cancer. Galectin-8 induces neutrophil adhesion and binding to vascular endothelial cells, and is involved in trans-endothelial migration [31]. Galectin-9 suppresses the generation of Th17 lymphocytes, induces early apoptosis, and promotes the generation of regulatory T cells in mouse collagen-induced arthritis. Galectin-9 also binds immunoglobulin (Ig) E and prevents the formation of allergen-IgE complex, thus precluding the release of chemical mediators from mast cells [32]. 

Among the galectin members, galectin-10 is predominantly distributed in human eosinophils and is the main component protein of Charcot–Leyden crystals (CLCs), which have been described in various eosinophilic diseases [33]. CLC-forming protein was initially identified as an eosinophil lysophospholipase, but in 1995, this protein was identified as galectin-10 [34]. CLCs are occasionally found in macrophages [35], but they are likely due to phagocytosis of other cell-derived CLCs [36,37,38]. Previous studies have reported that galectin-10 is present in some T cell subsets. In patients with atopic dermatitis, an elevation in galectin-10 expression is found in interleukin (IL)-22-producing T cells [39]. Although CD25+ regulatory T cells were shown to express galectin-10 [40], a recent report showed no significant galectin-10 mRNA and protein expression using multiple experimental modalities [41]. Galectin-10 and CLCs are also found in basophils [42,43], whereas the presence of CLCs in clinical samples is almost consistently associated with eosinophilic diseases [44]. This finding is likely due to the small number of basophils and their much smaller degree of local accumulation than eosinophils. 

Galectin-10 is the fifth most abundant protein in peripheral blood eosinophils, representing 7–10% of their total cytoplasmic protein [45]. In a study using two-dimensional gel electrophoresis for human peripheral eosinophil protein expression, galectin-10 was observed as streaks, owing to its insolubility, and it was the second most abundant protein [46]. Galectin-10 is a predominant, constitutively expressed eosinophil protein. The intracellular localization of CLC protein/galectin-10 in eosinophils has been studied since the 1980s. In 1988, Dvorak et al. reported that CLC protein was present in a minor population of cytoplasmic large vacuoles from mature eosinophils considered coreless granules, which were termed “primary granules”, and also in granule-poor cytoplasmic areas [47]. CLC protein was also found in the nucleus and cytoplasm of IL-5-stimulated eosinophils [48]. In 1997, Calafat et al. reported that CLC protein was localized in the cytoplasm and euchromatin of eosinophils and basophils, and that this protein increased in parallel with eosinophilic maturation [37]. However, in these earlier ultrastructural studies, the accurate intracellular localization remains incompletely understood, which could be because of technical limitations. Whether galectin-10, which is similar to other preformed eosinophil proteins, is stored within secretory granules has remained unclear. Moreover, in the light of current knowledge of eosinophil biology, human eosinophils do not contain primary granules, but only one population of secretory granules termed specific granules, which mature during the development of eosinophils [49].

The distribution of intracellular galectin-10 was recently assessed by immunostaining analysis and pre-embedding immunonanogold electron microscopy [50]. This technique enables precise intracellular detection and imaging of proteins at high resolution [51]. By applying several strategies, such as better antigen preservation, robust blocking of non-specific immunolabeling, and the use of small gold particles, studies have shown that galectin-10 is not present in eosinophil secretory granules, but in the peripheral cytoplasm of mature eosinophils [36,50]. Galectin-10 is mostly accumulated within an area of approximately 250 nm from the plasma membrane (Figure 1). Remarkably, galectin-10 is also able to interact with the plasma membrane, forming high-density microdomains throughout the entire extent of this membrane [50]. Moreover, the plasma membrane–galectin-10 interrelationship has been shown by the formation of galectin-10-positive plasma membrane-derived vacuoles upon the stimulation of endocytosis [49]. These endocytic vacuoles were probably formerly interpreted as primary granule sites for CLC protein [50].

The identification of galectin-10 as cytosolic pools within eosinophils is in accordance with other members of the galectin family. Other members of this family are synthesized as cytosolic proteins, and their externalization is not mediated through classical secretion, bypassing the Golgi complex [30]. The peripheral localization of galectin-10 in the cytoplasm of eosinophils and its strong association with the plasma membrane facilitate the release of galectin-10 through plasma membrane disintegration, as discussed below.

## 4. Release of Galectin-10 Is Not Mediated through Secretory Systems

Eosinophil granule-stored proteins can be released quickly without the process of de novo protein synthesis [49]. Live eosinophils release granular proteins through piecemeal degranulation or exocytosis [52]. Piecemeal degranulation differentially releases granule proteins, which are transported by discrete, large secretory vesicles [52]. In contrast, exocytotic degranulation releases the entire granule contents, wherein intracellular granules fuse with the plasma membrane, but this secretory process is not frequently found in vivo. Eosinophils also release their total cellular content through so-called cytolysis, which is distinct from apoptosis and necrosis. Cytolysis in inflamed tissues is associated with deposition of granular proteins and occurs in 30–80% of eosinophils [53,54,55,56]. Recent evidence has shown that cytolysis is an active cell death program that releases a net-like chromatin structure (i.e., extracellular trap cell death [ETosis]) [36,57,58,59].

In the in vitro experimental setting, human eosinophils undergo cytolysis/ETosis by various stimuli, such as calcium ionophore, phorbol 12-myristate 13-acetate, IL-5/granulocyte macrophage colony-stimulating factor with platelet-activating factor, immobilized or aggregated Igs, autoantibodies, and fungi [57,60,61,62,63,64]. Eosinophil ETosis (EETosis) has been implicated in a wide range of eosinophilic diseases, such as asthma [65,66], allergic bronchopulmonary aspergillosis [67,68], eosinophilic otitis [61,69], eosinophilic chronic rhinosinusitis (chronic rhinosinusitis with nasal polyps) [61], hypereosinophilic syndrome [36,57,70], Wells syndrome (eosinophilic cellulitis) [71], eosinophilic granulomatosis with polyangiitis ([EGPA] Churg–Strauss syndrome) [72,73], eosinophilic pneumonia [74,75,76], ulcerative colitis [59], and chronic obstructive pulmonary diseases [77]. 

Galectins may participate in liquid–liquid phase separated structures at the cell surface or intracellular locations [30]. As stated above, galectin-10 has a peculiar distribution in the eosinophil peripheral cytoplasm near the plasma membrane without compartmentalization in apparent secretory organelles. Therefore, extracellular release of galectin-10 is not mediated through conventional secretory process (i.e., piecemeal degranulation or exocytosis). Indeed, human eosinophils stimulated with C-chemokine ligand 11 or tumor necrosis factor-α, which are inducers of eosinophil secretion, do not change the cellular localization of galectin-10 [50]. Eosinophils undergoing ETosis rapidly disintegrate their plasma membrane to release large pools of galectin-10 [72]. 

During the temporal course of ETosis, eosinophils bud off plasma membrane-enveloped extracellular vesicles that retain galectin-10, even after rupture of the originating cell [36,57]. These extracellular vesicles are approximately 200–5000 nm in diameter, and can be deposited into the tissue because they do not expose phosphatidylserine [57], which is characteristic of phagocyte engulfment. Another interesting aspect of the EETosis process is that the subcellular localization of galectin-10 can change rapidly in that it crystallizes to form CLCs in the cytoplasm [36]. Furthermore, if EETosis occurs at the same time as a localized overaccumulation of eosinophils, the concentration of released galectin-10 is temporarily elevated, causing further extracellular crystallization [36]. The eosinophils isolated from asthmatic patients cultured under a hypoxic condition for more than 24 h induce CLC formation, although the detailed mechanism is unknown [78].

## 5. Detection of Cytolytic ETosis Using Immunohistochemical Staining

Pathological studies of patients with allergies or eosinophilic diseases have frequently shown abundant eosinophilic granule deposition, even in the absence of intact eosinophil infiltration [53]. Our laboratory has performed tissue immunostaining for galectin-10 and MBP in various diseases. Figure 2 shows typical images of nasal-polyp tissue from chronic rhinosinusitis with nasal polyps. Figure 2A shows intact eosinophils that were stained for cytoplasmic galectin-10 and MBP. In contrast, ETotic cells release the majority of soluble galectin-10 protein that can diffuse in the tissue (Figure 2B). Secreted small membrane-enclosed vesicles and CLCs that deposit in tissue are stained with anti-galectin-10 antibodies. Because eosinophil granules remain intact after the cell has disintegrated, MBP stains granularly [36,57]. MBP can be further deposited in the tissue after free granules have disintegrated because tissue persistence of MBP in tissue can be observed up to 6 weeks in vivo [79]. How long the released CLCs remain stable and detectable in tissue or secretions is not well understood, although it may vary depending on the microenvironment. An example of this variation is that a stable structure, such as a mucus plug fitted into a bronchus or sinus cavity, might be hardly phagocytosed by macrophages. Therefore, the crystal structure is expected to remain for a long time, probably for weeks.

The MBP and galectin-10 double-staining method highlights the difference between intact and ETotic cells. To assess them quantitatively, we established a cytolysis index calculated as the MBP/galectin-10 ratio [72]. In brief, binary images were obtained and each stained area was measured by imaging software. If cells undergo ETosis, the loss of galectin-10 results in an increased MBP/galectin-10 ratio. In a histopathological examination of various affected tissues from patients with EGPA, the MBP/galectin-10 ratio exceeded the reference value in 22 of 23 specimens [72]. Although the presence of CLC and extracellular vesicles may potentially hinder an accurate evaluation of the cytolysis index, they have a consistently smaller area than the extracellular deposition of MBP in a low-magnification image (Figure 2(Bi)).

## 6. Clinical Significance of Galectin-10 as a Clinical Biomarker

Galectin-10 has been recognized as a promising biomarker for several eosinophilic diseases (Table 1), either alone or in combination with other biomarkers. Galectin-10 can be measured from sputum, nasal secretions, nasal polyps, serum, and skin tissue, depending on the disease. Galectin-10 was identified as a six-gene expression biomarker in sputum samples that discriminates the inflammatory phenotypes of asthma and predicts the inhaled-corticosteroid treatment response [80]. In another report, eight genes (CLC, EMR4P, IL5RA, FRRS1, HRH4, SLC29A1, SIGLEC8, IL1RL1) were consistently overexpressed in all types of multimorbidity for asthma, dermatitis, and rhinitis [81]. Galectin-10 has been identified as one of the most frequently expressed genes in the sputum transcriptome from severe asthma [82]. Protein expression of galectin-10 in sputum has a strong correlation with the sputum eosinophil percentage, and sputum galectin-10 concentrations have a higher level of diagnostic accuracy than the sputum eosinophil percentage [83]. Serum [84] or sputum [85] galectin-10 concentrations in patients with asthma were shown to be significantly decreased after the administration of the anti-IL-5 humanized monoclonal antibody mepolizumab. 

In eosinophilic esophagitis, an electron microscopy study showed that approximately 80% of eosinophils underwent cytolysis [55]. Furuta et al [86]. and Ackerman et al [87]. measured galectin-10 concentrations in esophageal luminal samples from patients with eosinophilic esophagitis and found that these concentrations correlated well with eosinophilic inflammation in tissue. Plasma biomarkers including galectin-10 have been reported as noninvasive biomarkers in the diagnosis of eosinophilic esophagitis. A panel comprising eosinophil-associated proteins along with the absolute eosinophil count are superior to the eosinophil count alone, in distinguishing eosinophilic esophagitis from controls [88]. The authors suggested measuring plasma galectin-10 rather than serum, since galectin-10 might increase in serum as a consequence of blood clotting [88].

The presence of CLCs in nasal-polyp tissue is associated with the disease severity and IL-5 levels [36,86]. A study showed that galectin-10 concentrations in nasal secretions from patients with chronic rhinosinusitis with nasal polyps were a noninvasive biomarker that predicted a better response to glucocorticosteroids [87]. In this study, patient groups were divided into responders and non-responders according to the nasal polyp score after 2 weeks of oral glucocorticosteroid treatment. The average concentration of galectin-10 before glucocorticosteroid treatment was 160 times higher in the responder group than in the non-responder group [87]. Another research group reported that the baseline relative CLC mRNA level in nasal brushing samples was higher in patients with disease recurrence than in those without recurrence [89]. In nasal-polyp tissue, galectin-10 protein expression levels, but not the eosinophil count of tissue section, are positively correlated with tissue *Aspergillus fumigatus* antigen levels [90].

**Table 1 biomolecules-12-01385-t001:** Galectin-10 as a biomarker in human diseases.

Disease	Sample	Methods	Purpose/Outcome	Reference
Aspirin-induced asthma	Serum	qRT-PCR	Distinguishing AIA from ATA	[91]
Asthma with or without ABPA	Sputum	Western blot	Correlation with the sputum eosinophil count	[83]
Asthma	Sputum	Inhouse ELISA	Correlation with the sputum eosinophil counts	[92]
Severe asthma	Sputum	RNA transcriptome	The most upregulated gene in severe asthma	[82]
Eosinophilic asthma	Serum	Commercial ELISA	Monitoring after mepolizumab treatment	[84]
Eosinophilic esophagitis	Plasma	Inhouse ELISA	Distinguishing EoE	[88]
Eosinophilic esophagitis	Luminal secretions (esophageal string test)	Inhouse ELISA	Diagnosis of EoE in children	[93]
Eosinophilic esophagitis	Luminal secretions (esophageal string test)	Inhouse ELISA	Distinguishing active EoE in children and adult	[94]
CRSwNP	Nasal secretions	Commercial ELISA	Predicting the glucocorticoid response	[87]
CRSwNP	Nasal brushing	qRT-PCR	Predictive factor for recurrence	[89]
CRSwNP	Nasal polyps	Western blot, qRT-PCR, IHC	Predictive factor for the phenotype	[95]
CRSwNP	Nasal polyps	Commercial ELISA	Correlation with Asp f 1	[90]
CRSwNP	Nasal secretions	IHC	Correlation with severity	[96]
Nonallergic rhinitis with eosinophilia syndrome	Nasal secretions	ELISA	Correlation with severity and treatment	[97]
Atopic dermatitis	Skin specimens, serum	IHC, commercial ELISA	Diagnosis, correlation with severity	[39]
Eosinophilic granulomatosis with polyangiitis	Serum	Commercial ELISA	Correlation with disease activity	[72]

Abbreviations: ABPA, allergic bronchopulmonary aspergillosis; CRSwNP, chronic rhinosinusitis with nasal polyps; qRT-PCR, quantitative real-time polymerase chain reaction; ELISA, enzyme-linked immunosorbent assay; IHC, immunohistochemistry; AIA, aspirin-induced asthma; ATA, aspirin-tolerant asthma; EoE, eosinophilic esophagitis.

With regard to systemic vasculitis, we have reported that serum eosinophil granule proteins and galectin-10 concentrations were increased in EGPA and positively correlated with disease activity, as assessed by the Birmingham Vasculitis Activity Score [72]. When normalized with the blood eosinophil count, this correlation remained for galectin-10, but not for granule proteins, including EDN and eosinophil cationic protein. In patients with active EGPA, serum IL-5 concentrations positively correlated with galectin-10 concentrations, but not with serum EDN concentrations, eosinophil cationic protein concentrations, and the blood eosinophil count [72]. Therefore, serum galectin-10 concentrations are a unique biomarker that might reflect the systemic occurrence of cytolytic ETosis. The release of galectin-10 and membrane-bound, cell-free intact eosinophil granules from EETotic eosinophils might be associated with differing serum concentrations of galectin-10 and eosinophil granule proteins. In patients with EGPA, eosinophils undergoing ETosis in small vessels potentially contribute to the increase in serum galectin-10 concentrations and lead to thrombus formation by providing a scaffold for platelets in addition to promoting vascular injury [73].

## 7. Biological Functions of CLCs and Galectin-10

Until recently, the functional role of CLC/galectin-10 in allergic diseases remained unknown. To date, several studies have shown that CLCs are more than just markers of eosinophilic inflammation and have a functional role in immunity. Persson et al. produced recombinant galectin-10 crystals that were structurally and biochemically similar to CLCs and injected them into the airways of naïve mice [86]. Galectin-10 crystals induced an innate immune response, which was rich in neutrophils and monocytes, and led to the uptake of crystals by dendritic cells. CLC-dissolving antibodies suppressed airway inflammation, goblet-cell metaplasia, bronchial hyperreactivity, and IgE synthesis induced by CLCs, thus suggesting a possible therapeutic approach [86].

Another study provided evidence that CLCs are recognized by the NLRP3 inflammasome and induce the release of the proinflammatory cytokine IL-1β after uptake by human macrophages in vitro [98]. This study also showed that the administration of CLCs into the lungs of mice resulted in inflammasome activation and an increase in IL-1β concentrations in bronchoalveolar lavage fluid. In addition, CLCs cause epithelial cells to recruit neutrophils and produce cytokines that prime neutrophil functions [99]. Neutrophils induce NETosis after stimulation with CLCs, although the effect of CLCs on eosinophil EETosis is not obvious [99]. These results support the concept that crystals can trigger cell damage through an auto-amplification loop involving the release of damage-associated molecular patterns called “crystallopathies” [100]. 

Several studies have shown intracellular functions for galectin-10. Galectin-10 interacts with human eosinophil granule cationic ribonucleases and contributes to their intracellular packaging [101]. Galectin-10 in CD25+ regulatory T cells contributes to their proliferative capacity and suppressive effects [40]. However, a recent rebuttal report indicated that the number of galectin-10 regulatory T cells was low, and galectin-10 had no effect on regulatory T cell viability, proliferation, or expression of their master transcription factor FOXP3 [41].

## 8. Lessons from Eosinophilic Gastrointestinal Disorders

Since physiological eosinophils with remarkable degranulation are exclusively present in the gastrointestinal tract except for the esophagus, the diagnosis of non-eosinophilic esophagitis eosinophilic gastrointestinal disorders requires supporting clinical findings as well as digestive eosinophil counts [102]. Histopathological research of digestive eosinophil counts has become advanced, resulting in novel criteria (e.g., Pesek’s criteria corresponding to clinical practice) [103]. Nevertheless, some issues still need to be addressed. One issue is that the difference between resident degranulated eosinophils in the gut and inflammatory degranulated eosinophils that damage tissue leading to dysfunction is unknown. Whether the presence of eosinophils is always seen in eosinophilic inflammation needs to be determined. Interestingly, a report addressed this issue in patients with eosinophilic esophagitis. There was a discrepancy between the eosinophil count and the deposition of extracellular EDN in the tissue of eosinophilic esophagitis [27]. This finding suggests that the tissue eosinophil count occasionally underestimates how extensively eosinophils are involved, suggesting that the deposition of extracellular EDN has been left after the disappearance of eosinophils. Interesting findings have been shown in patients with infantile food protein-induced allergic proctocolitis, which is basically synonymous with infantile eosinophilic colitis. The patients with food protein-induced allergic proctocolitis showed not only severe eosinophilic infiltration with degranulation, but also CLCs [104]. CLCs are generated when eosinophils fall into EETosis, which causes dysregulation of galectin-10, and this is associated with the severity of inflammation. Surprisingly, despite the detection of CLCs, food protein-induced allergic proctocolitis generally shows bloody stool, but this is otherwise healthy, suggesting a good prognosis [105]. Indeed, breastfed infants can be improved spontaneously, and the resolution of symptoms is usually observed in a few days after avoidance of offending proteins such as cow’s milk. Subsequently, most infants become tolerant to the causative food by 3 years of age at the latest. Further examination of the detailed involvement of tissue eosinophils needs to be performed to determine whether discrepancies between the tissue eosinophil count, deposition of extracellular EDN or detection of CLCs, and the disease severity or degree of eosinophilic inflammation are exclusively observed in the gastrointestinal tract. Comprehensive analyses by evaluating eosinophil activation, eosinophils, and EETosis might be useful for assessing eosinophil inflammation.

## 9. Myeloproliferative Diseases and CLCs

The potential of galectin-10 as a biomarker includes its evaluation in myeloproliferative diseases. CLCs have been reported in the tissue and bone marrow from patients with hypereosinophilic syndrome, including eosinophilic leukemia [70,106,107,108,109,110]. Indeed, CLCs were originally identified in the cardiac blood and spleen in patients who died from leukemia [111]. In addition, other myeloproliferative diseases cause CLC formation [112]. CLCs can be produced from whole-cell lysates of AML14.3D10 cells, which are a subclone of the AML14 cell line that was established from a patient with FAB M2 acute myeloid leukemia in the experimental setting [98,113]. 

We searched for the occurrence of CLC in myeloproliferative diseases that were not classified as hypereosinophilic syndrome. Twelve cases have been described in the literature to date, with nine cases of acute myeloid leukemia, one case of myeloid neoplasm, one case of chronic myeloid leukemia, and one case of T cell lymphoblastic lymphoma (Table 2). Bone-marrow aspiration and trephine biopsy were performed in all of these cases, and bone-marrow necrosis with CLCs was shown in most of them. These findings strongly suggest that, in some myeloproliferative diseases, CLCs are formed by an increase in the number of galectin-10-expressing cells and their necrosis. Gene expression and protein concentrations of galectin-10 in leukemia have not been fully investigated. Further studies are required on their roles in pathogenesis and as a disease biomarker.

## 10. Conclusions

Except for some leukemia cells, basophils, and some T cell populations, galectin-10 is abundantly and exclusively present in eosinophils in the human body. Unlike eosinophil granule proteins, galectin-10 is not present in secretory granules and is not released by the conventional degranulation mechanism. Eosinophil cell death in inflammatory tissues is rarely accidental necrosis, and mostly ETosis [44,59]. Apoptotic eosinophils are rarely observed in tissues, which may be due to a rapid phagocytic clearance (efferocytosis) [54]. Efficient efferocytosis spares the tissue microenvironment from the harmful effects of decomposing cells [126]. Compared with the highly cationic granule proteins that strongly bind to negatively-charged tissue elements and aggregate with long half-lives [127,128,129], galectin-10 might be a unique marker with different characteristics. The increase in galectin-10 protein concentrations in body fluids could be a surrogate biomarker of a potentially proinflammatory type of eosinophil cytolytic cell death (Figure 3). Future studies are required to determine its clinical significance in various eosinophil-related diseases.

## Figures and Tables

**Figure 1 biomolecules-12-01385-f001:**
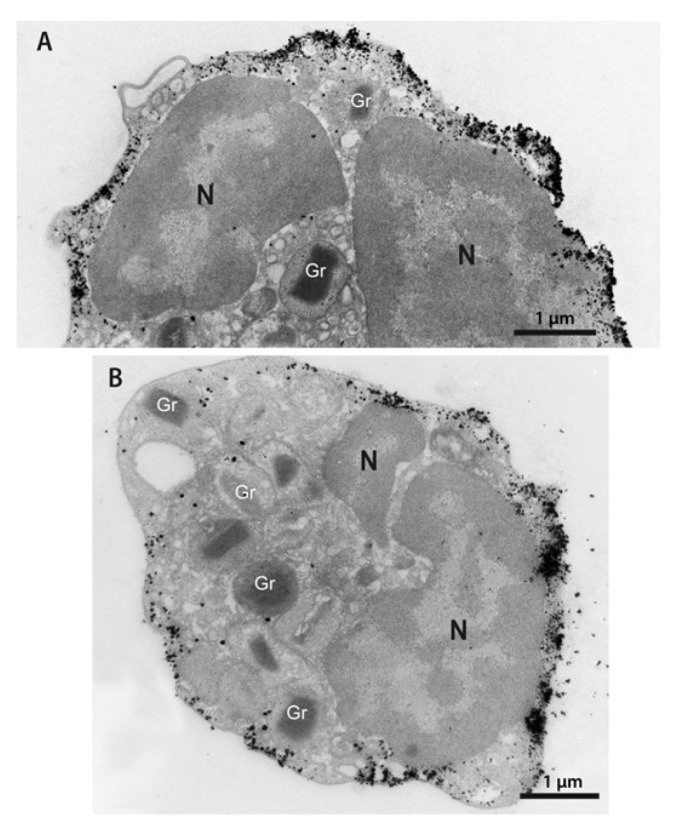
(**A**,**B**) Distribution of galectin-10 in the eosinophil cytoplasm. Note a robust pool of this protein as show by the deposition of gold particles in the peripheral cytoplasm and in association with the plasma membrane. Secretory granules (Gr) are negative. Isolated human eosinophils were processed for immunonanogold transmission electron microscopy for galectin-10, as previously described [50]. N, nucleus.

**Figure 2 biomolecules-12-01385-f002:**
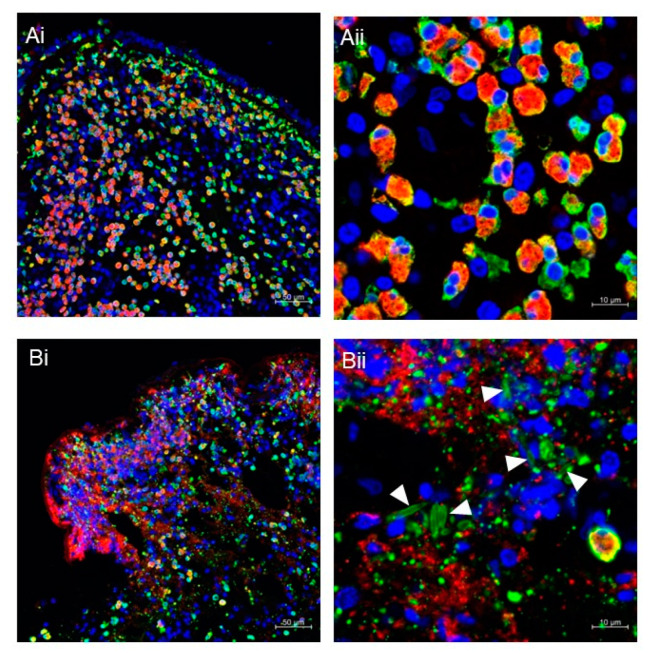
(**A**,**B**) Immunofluorescence staining of nasal polyps from patients with chronic rhinosinusitis with nasal polyps. Fluorescence images for MBP (red), galectin-10 (green), and DNA (blue) were obtained using a Carl Zeiss LSM780 confocal microscope, as previously described [36]. Intact eosinophils are massively infiltrated in polyps (**Ai**,**Aii**). The presence of ETotic eosinophils with extracellular traps (degenerated DNA), cell-free granules (punctate MBP), extracellular vesicles (punctate galectin-10), and CLCs (galectin-10-stained needle-like crystals; arrowheads) can be seen (**Bi**,**Bii**). Scale bars: 50 μm for (**i**) and 10 μm for (**ii**).

**Figure 3 biomolecules-12-01385-f003:**
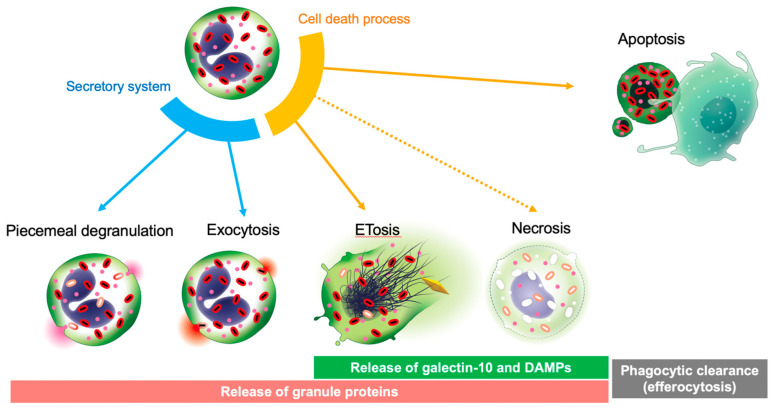
ETosis-mediated galectin-10 release. Live eosinophils secrete granule proteins through piecemeal degranulation and exocytosis. Eosinophils have at least two distinct active cell death pathways of apoptosis and ETosis. Unlike apoptotic cells, which are engulfed and digested by phagocytes (efferocytosis), eosinophils can undergo ETosis in tissue and release their cellular content, including cytoplasmic galectin-10, entire secretory granules, and damage-associated molecular patterns. Therefore, ETosis is also considered a form of eosinophil secretion.

**Table 2 biomolecules-12-01385-t002:** Cases related to leukemia with CLC accumulation reported in the literature.

Diagnosis	Sex/Age (y)	Tissue Findings	Disease Status	Reference
AML	F/45	BM necrosis	Not described	[114]
AML	M/66	BM necrosis	Remission	[115]
MDS/MPN transformed into AML	F/60	BM extensive necrosis	Not described	[116]
AML	M/59	BM necrosis	Remission	[117]
AML	F/70	BM extensive necrosis	Died	[118]
AML with mutated NPM1	F/51	BM mostly necrosis	Remission	[119]
AML with mutated NPM1	M/43	BM necrotic hematopoietic tissue	Remission	[120]
MDS transformed into AML	M/82	BM necrosis	Died	[121]
AML with inv (16)	M/35	Hypercellular BM	Not described	[122]
Myeloid neoplasm with mutated NPM1 and TET2	M/46	BM necrosis	Remission	[123]
Chronic myelogenous leukemia	M/65	BM necrosis	Not described	[124]
T cell lymphoblastic lymphoma	M/49	Focal cell necrosis in BM	Not described	[125]

Abbreviations: CLC, Charcot–Leyden crystal; F, female; M, male; AML, acute myeloid leukemia; MDS/MPN, myelodysplastic/ myeloproliferative neoplasm.

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
