# Peer review of "Galectin-10 as a Potential Biomarker for Eosinophilic Diseases"

_biomolecules, 2022, doi:10.3390/biom12101385_

Round 1
Reviewer 1 Report
This manuscript is an excellent review article. I have some comments.
Line 79.
Is the phrase “and are constitutively observed” meaningful? I guess the phrase may not be necessary.
Line 121.
The word “remains” may need to be changed to “have remained”, in order to coincide with the following paragraphs.
Line 179.
C-chemokine needs to be changed to CC-chemokine.
References.
The names of journals need to be written following the instruction (references 12, 28, 61, 67 and 69).
Reviewer 2 Report
General Comments:
This is a thorough, clearly written and comprehensive review of galectin-10/Charcot-Leyden Crystal protein as a potential biomarker in eosinophil-associated diseases. The citations are up-to-date and cover the state-of-the-art in terms of studies identifying and characterizing Gal-10/CLC as a blood, tissue and body-fluid biomarker of eosinophil activation/cell death,m principally from the ETosis cell death pathway. The review could benefit from some editing of the English as a number of sentences are difficult to understand due to problems in sentence structure or missing information as noted specifically below.
Specific Comments:
1. Line 53: The authors should mention both MBP-1 and MBP-2, both of which are components of the eosinophil's secondary granule.
2. Line 55: "...the concentration of these biomarkers..." In what? Blood, tissues, secretions etc. The authors should specify in what.
3. Line 66: ..."severe diseases...(add "in asthma")
4. Line 82: "Galectin is..." This should read "Galectins are...a family of..."
5. Line 245: The authors should also cite the following reference as new #88 and in Table 1 that quantified Galectin-10/CLC protein in luminal secretions captured by a 1-hr esophageal string test in patients with EoE, and line 243 should read "Furuta et al and Ackerman et al..."
Ackerman SJ, Kagalwalla AF, Hirano I, Gonsalves N, Menard Katcher P, Gupta S, Wechsler JB, Grozdanovic M, Pan Z, Masterson JC, Du J, Fantus RJ, Alumkal P, Lee JJ, Ochkur S, Ahmed F, Capocelli K, Melin-Aldana H, Biette K, Dubner A, Amsden K, Keeley K, Sulkowski M, Zalewski A, Atkins D, Furuta GT. The 1-Hour Esophageal String Test: A Non-Endoscopic Minimally Invasive Test that Accurately Detects Disease Activity in Eosinophilic Esophagitis. Amer. J. Gastroenterol. 2019;114:1614-1625.
6. The authors should mention that Galectin-10/CLC is increased in serum as a consequence of blood clotting, and may therefore be a mere surrogate of the number of eosinophil's in the blood, and it should therefore be measured in plasma, not serum, as described in reference #88 - Wechsler et al Allergy, in the manuscript.
7. Lines 311-312: This sentence is unclear in terms of "should be required..." and needs revision. What should be required?
